# A Randomized Algorithm to Reduce the Support of Discrete Measures

**Francesco Cosentino**
Mathematical Institute
University of Oxford
The Alan Turing Institute
francesco.cosentino@maths.ox.ac.uk

**Harald Oberhauser**
Mathematical Institute
University of Oxford
oberhauser@maths.ox.ac.uk

**Alessandro Abate**
Dept. of Computer Science
University of Oxford
The Alan Turing Institute
alessandro.abate@cs.ox.ac.uk

## Abstract

Given a discrete probability measure supported on $N$ atoms and a set of $n$ real-valued functions, there exists a probability measure that is supported on a subset of $n+1$ of the original $N$ atoms and has the same mean when integrated against each of the $n$ functions. If $N \gg n$ this results in a huge reduction of complexity. We give a simple geometric characterization of barycenters via negative cones and derive a randomized algorithm that computes this new measure by "greedy geometric sampling". We then study its properties, and benchmark it on synthetic and real-world data to show that it can be very beneficial in the $N \gg n$ regime. A Python implementation is available at https://github.com/FraCose/Recombination_Random_Algos.

## 1   Introduction

Discrete probability measures are central to many inference tasks, for example as empirical measures. In the "big data" regime, where the number $N$ of samples is huge, this often requires to construct a reduced summary of the original measure. Often this summary is constructed by sampling $n$ points at random out of the $N$ points, but Tchakaloff's theorem suggests that there is another way.

**Theorem 1** (Tchakaloff [1]). *Let $\mu$ be a discrete probability measure that is supported on $N$ points in a space $\mathscr{X}$. Let $\{f_1, \ldots, f_n\}$ be a set of $n$ real-valued functions $f_i : \mathscr{X} \to \mathbb{R}$, $n < N$. There exists a discrete probability measure $\hat{\mu}$ such that* $\mathrm{supp}(\hat{\mu}) \subset \mathrm{supp}(\mu)$, $|\mathrm{supp}(\hat{\mu})| \leq n+1$, *and*

$$\mathbb{E}_{X \sim \mu}[f_i(X)] = \mathbb{E}_{X \sim \hat{\mu}}[f_i(X)] \text{ for all } i \in \{1, \ldots, n\}. \tag{1}$$

We introduce a randomized algorithm that computes $\hat{\mu}$ efficiently in the $n \ll N$ regime.

**Related work.**   Reducing the support of a (not necessarily discrete) measure subject to matching the mean on a set of functions is a classical problem, which goes back at least to Gauss' famous quadrature formula that matches the mean of monomials up to a given degree when $\mu$ is the Lebesgue measure on $\mathbb{R}$. In multi-dimensions this is known as cubature and Tchakaloff [1] showed the existence of a reduced measure for compactly supported, not necessarily discrete, measures, see [2]. When $\mu$ is discrete, the problem of computing $\hat{\mu}$ also runs under the name recombination. Algorithms to compute $\hat{\mu}$ for discrete measures $\mu$ go back at least to [3] and have been an intensive research topics ever since; we refer to [4] for an overview of the different approaches, and [4, 5, 6] for recent, state of the art algorithms and applications. Using randomness for Tchakaloff's Theorem has been suggested before [7, 8] but the focus there is to show that the barycenter lies in the convex hull when enough points from a continuous measure are sampled so that subsequently any of the algorithms [4, 5, 6] can be applied; in contrast, our randomness stems from the reduction algorithm itself. More generally, the topic of replacing a large data set by a small, carefully weighted subset is a vast field that has

attracted many different communities and we mention, pars pro toto, computational geometry [9], coresets in computer science [10], scalable Bayesian statistics [11], clustering and optimisation [12]. In follow up work [13], our randomized algorithm was already used to derive a novel approach to stochastic gradient descent.

**Contribution.** The above mentioned algorithms [4, 5, 6] use a divide and conquer approach that splits up points into groups, computes a barycenter for each group, and solves a constrained linear system several times. This leads to a deterministic complexity that is determined by $N$ and $n$. In contrast, our approach uses the geometry of cones to "greedy" sample for candidates in the support of $\mu$ that are atoms for $\hat{\mu}$ and tries to construct the reduced measure in one go. Further, it can be optimized with classical black box reset strategies to reduce the variance of the run time. Our results show that this can be very efficient in the big data regime $N \gg n$ that is common in machine learning applications, such as least least square solvers when the number of samples $N$ is very large. Moreover, our approach is complementary to previous work since it can be combined with it: by limiting the iterations for our randomized algorithm and subsequently running any of the deterministic algorithms above if a solution by "greedy geometric sampling" was not found, one gets a hybrid algorithm that is of the same order as the deterministic one but that has a good chance of being faster; we give full details in Appendix E but focus in main text on the properties of the randomized algorithm.

**Outline.** We introduce the basic ideas in Section 2, where we derive a simple version of the greedy sampling algorithm, and study its theoretical properties. In Section 3 we optimize the algorithm to better use the cone geometry, combine with reset strategies to reduce the running time variance, and use the Woodbury formula to obtain a robustness result. In Section 4 we discuss numerical experiments that study the properties of the algorithms on two problems: (i) reducing the support of empirical measures; and (ii) least square solvers for large samples. In the Appendix we provide detailed proofs and more background on discrete geometry.

## 2 Negative cones and a naive algorithm

**Background.** As is well-known, Theorem 1 follows from Caratheodory's convex hull theorem

**Theorem 2** (Caratheodory). *Given a set of $N$ points in $\mathbb{R}^n$ and a point $x$ that lies in the convex hull of these $N$ points. Then $x$ is a linear combination of at most $n+1$ points from the $N$ points.*

It is instructive to recall how Theorem 2 implies Theorem 1. Therefore define a $\mathbb{R}^n$-valued random variable $F : \Omega = \mathscr{X} \to \mathbb{R}^n$ as $F(\omega) := (f_1(\omega), \dots, f_n(\omega))$ and note that Equation (1) is equivalent to

$$\int_\Omega F(\omega)\mu(d\omega) = \int_\Omega F(\omega)\hat{\mu}(d\omega).$$

Since $\mu$ has finite support, the left-hand side can be written as a sum $\sum_{\omega \in \mathrm{supp}(\mu)} F(\omega)\mu(\omega)$. This sum gives a point in the convex hull of the set of $N$ (or less) points $\mathbf{x} := \{F(\omega) : \omega \in \mathrm{supp}(\mu)\}$ in $\mathbb{R}^n$. But by Caratheodory's theorem, this point must be a convex combination of a subset $\hat{\mathbf{x}}$ of only $n+1$ (or less) points of $\mathbf{x}$ and Theorem 1 follows. This proof of Theorem 1 is also constructive in the sense that it shows that computing $\hat{\mu}$ reduces to constructing the linear combination guaranteed by Caratheodory's theorem; e.g. by solving $N$ times a constrained linear system, see [3].

**Barycenters and cones.** Key to Tchakaloff's theorem is to verify if two measures have the same mean. We now give a simple geometric characterization in terms of negative cones, Theorem 3.

**Definition 1.** *Let $\mathbf{x} \subset \mathbb{R}^n$ be a finite set of points in $\mathbb{R}^n$. We call the set $C(\mathbf{x}) := \{c \in \mathbb{R}^n \,|\, c = \sum_{x \in \mathbf{x}} \lambda_x x,$ where $\lambda_x \geq 0\}$ the cone generated by $\mathbf{x}$ and we also refer to $\mathbf{x}$ as its basis. We call the set $C^-(\mathbf{x}) := \{c \,|\, c = \sum_{x \in \mathbf{x}} \lambda_x x,$ where $\lambda_x \leq 0\}$ the negative cone generated by $\mathbf{x}$.*

For example, $C(x_1, x_2)$ is the "infinite" triangle created by the half-lines $\overline{0x_1}$, $\overline{0x_2}$ with origin in 0; $C(\{x_1, x_2, x_3\})$, is the infinite pyramid formed by the edges $\overline{0x_1}$, $\overline{0x_2}$, $\overline{0x_3}$, with vertex 0; see Figure 1.

**Theorem 3.** *Let $\mathbf{x} = \{x_1, \dots, x_{n+1}\}$ be a set of $n+1$ points in $\mathbb{R}^n$ such that $\mathbf{x} \setminus \{x_{n+1}\}$ spans $\mathbb{R}^n$. Let $A$ be the matrix that transforms $\mathbf{x} \setminus \{x_{n+1}\} = \{x_1, \dots, x_n\}$ to the orthonormal basis $\{e_1, \dots, e_n\}$ of $\mathbb{R}^n$, i.e. $Ax_i = e_i$. Further, let $h_i$ be the unit vector such that $\langle h_i, x \rangle = 0$ for all $x \in \mathbf{x} \setminus \{x_i, x_{n+1}\}$ and $\langle h_i, x_i \rangle < 0$ and denote with $H_{\mathbf{x} \setminus \{x_{n+1}\}}$ a $n \times n$ matrix that has $h_1, \dots, h_n$ as row vectors. It holds that*

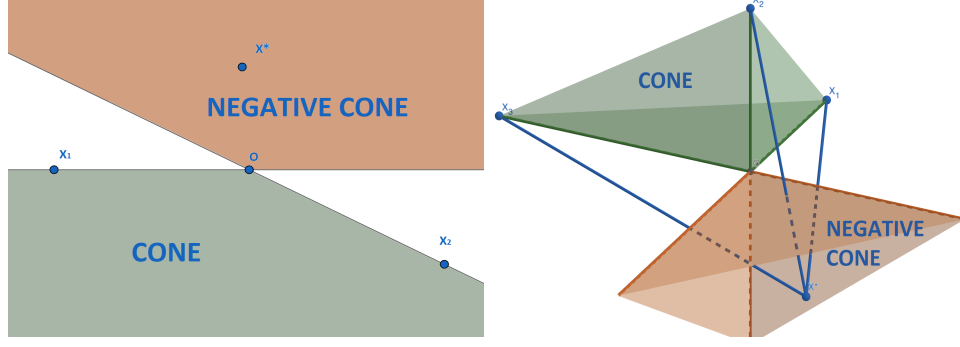

Figure 1: Cones and Negative Cones spanned by two (left) and three points (right).

1. $C(\mathbf{x} \setminus \{x_{n+1}\}) = \{c | H_{\mathbf{x} \setminus \{x_{n+1}\}} c \leq 0\}$ and $C^-(\mathbf{x} \setminus \{x_{n+1}\}) = \{c | H_{\mathbf{x} \setminus \{x_{n+1}\}} c \geq 0\}$.

2. $Ax \geq 0$ if and only if $H_{\mathbf{x} \setminus \{x_{n+1}\}} x \leq 0$ and $Ax \leq 0$ if and only if $H_{\mathbf{x} \setminus \{x_{n+1}\}} x \geq 0$.

3. There exists a convex combination of $\mathbf{x}$ with $0$ as barycentre, $\sum_{i=1}^{n+1} w_i x_i = 0$ for some $w_i > 0$, and $\sum_{i=1}^{n+1} w_i = 1$ if and only if $x_{n+1} \in C^-(\mathbf{x} \setminus \{x_{n+1}\})$.

The above result could be formulated without the matrix $A$, only in terms of $H_{\mathbf{x} \setminus \{x_{n+1}\}}$. However, $A$ is the inverse of the matrix with columns equal to the vectors $\{\mathbf{x} \setminus \{x_{n+1}\}\}$, hence computing $A$ is more efficient than computing $H_{\mathbf{x} \setminus \{x_{n+1}\}}$, since matrix inversion is optimized in standard libraries.

**A Naïve Algorithm.** Theorem 3 implies a simple randomized algorithm: sample $n$ points at random until the negative cone spanned by the $n$ points is not empty. Then item 3 of Theorem 3 implies that the $n$ points in the cone and any point in the negative cone form the support a reduced measure. If $\mathbb{E}_\mu[X] \neq 0$ we can always study the points $\mathbf{x} - \mathbb{E}_\mu[X]$, this is equivalent in the proof of Theorem 3 to work with cones whose vertex is not $0$.

---

**Algorithm 1** Basic measure reduction algorithm

1: **procedure** REDUCE(A set $\mathbf{x}$ of $N$ points in $\mathbb{R}^n$)
2:      Choose $n$ points $\mathbf{x}^\star$ from $\mathbf{x}$
3:      **while** $C^-(\mathbf{x}^\star) \cap \mathbf{x} = \emptyset$ **do**
4:          Replace $\mathbf{x}^\star$ with $n$ new random points $\mathbf{x}^\star$ from $\mathbf{x}$
5:      **end while**
6:      $\mathbf{x}^\star \leftarrow \mathbf{x}^\star \cup x^\star$ with an arbitrary $x^\star \in C^-(\mathbf{x}^\star) \cap \mathbf{x}$
7:      Solve the linear system $\sum_{x \in \mathbf{x}^\star} w_x^\star x = 0$ for $w^\star = (w_x^\star)_{x \in \mathbf{x}^\star}$
8:      **return** $(\mathbf{x}^\star, w^\star)$
9: **end procedure**

---

**Corollary 1.** *Algorithm 1 computes a reduced measure $\hat{\mu}$ as required by Theorem 1 in $\tau \cdot O(n^3 + Nn^2)$ computational steps. Here $\tau = \inf\{i \geq 1 : C^-(\mathbf{X}_i) \cap \mathbf{x} \neq \emptyset\}$, where $\mathbf{X}_1, \mathbf{X}_2, \ldots$ are random sets of $n$ points sampled uniformly at random from $\mathbf{x}$.*

The complexity of Algorithm 1 in a single loop iteration is dominated by (i) computing the matrix $A$ that defines the cones $C^-(\mathbf{x}^\star)$ and $C(\mathbf{x}^\star)$, (ii) checking if there are points inside the cones, (iii) solving a linear system to compute the weights $w_i^\star$. Respectively, the worst case complexities are $O(n^3)$, $O(Nn^2)$ and $O(n^3)$, since to check if there are points inside the cones we have to multiply $A$ and $\mathbf{X}$, where $\mathbf{X}$ is the matrix whose rows are the vector in $\mathbf{x}$.

**Proposition 1.** *Let $N > n + 1$ and $\mu$ be a discrete probability with finite support and $f_1, \ldots, f_n$ be as in Theorem 1. Moreover wlog assume $\mathbb{E}_{X \sim \mu}[f_i(X)] = 0$ for $i = 1, \ldots, n$. With $p := \frac{n \cdot n! (N-n)!}{N!}$ it holds that $\mathbb{E}[\tau] \leq \frac{1}{p}$ and $\mathrm{Var}(\tau) \leq \frac{1-p}{p^2}$ and, for fixed $n$, $\lim_{N \to \infty} \mathbb{E}[\tau] = 1$.*

Not surprisingly, the worst case bound for $\mathbb{E}[\tau]$ are not practical, and it is easy to come up with examples where this $\tau$ will be very large with high probability, e.g. a cluster of points and one

point far apart from this cluster would result in wasteful oversampling of points from the cluster. Such examples, suggest that one should be able to do better by taking the geometry of points into account which we will do in Section 3. However, before that, it is instructive to better understand the properties of Algorithm 1 when applied to empirical measures.

**Application to empirical measures.** Consider a random probability measure $\mu = \frac{1}{N} \sum_{i=1}^{N} \delta_{(f_1(X_i),...,f_n(X_i))}$ where the $X_1, X_2, \ldots$ are independent and identically distributed.

**Proposition 2.** *Let $N > n+1$ and let $f_1, \ldots, f_n$ be $n$ real-valued functions and $X_1, \ldots, X_N$ be $N$ i.i.d. copies of a random variable $X$. Set $F(X) = (f_1(X), \ldots, f_n(X))$, assume $\mathbb{E}[F(X)] = 0$ and denote*

$$E := \{0 \in \mathrm{Conv}\{F(X_i), i \in \{1, \ldots, N\}\}\}. \tag{2}$$

*1. $\mathbb{E}[\tau|E] \leq \frac{1}{p}$ and $Var(\tau|E) \leq \frac{1-p}{p^2}$, where*

$$p = \max\left\{ \frac{n \cdot n!(N-n)!}{N!}, 1 - \mathbb{P}\left(0 \notin \mathrm{Conv}\{F(X_1), \ldots, F(X_{n+1})\}\right)^{N-n} \right\}, \tag{3}$$

*2. If the law of $F(X)$ is invariant under reflection in the origin, then $\mathbb{P}\left(0 \notin \mathrm{Conv}\{F(X_1), \ldots, F(X_{n+1})\}\right) = 1 - 2^{-n}$,*

*3. For fixed $n$, as $N \to \infty$*

$$\mathbb{P}(\textit{for } n \textit{ uniformly at random chosen points } \mathbf{x}^\star \textit{ from } \mathbf{x}, \exists x \in \mathbf{x} \textit{ s.t. } x \in C^-(\mathbf{x}^\star)) \to 1,$$

*where $\mathbf{x} = \{F(X_1), F(X_2), \ldots, F(X_N)\}$.*

Proposition 2 conditions on the event (2) so that the recombination problem is well-posed, but this happens with probability one for large enough $N$, see Theorem 5 in Appendix and [8]. Not surprisingly, the worst case bounds of Algorithm 1 can be inconvenient, as equation (3) shows. Nevertheless, item 2 of Proposition 2 shows an interesting trade-off in computational complexity, since the total cost

$$\mathbb{E}[\tau]O(n^3 + Nn^2) \leq C(n^3 + Nn^2) \min\left\{ \frac{N!}{n \cdot n!(N-n)!}, \frac{1}{1 - (1 - 2^{-n})^{N-n}} \right\} \tag{4}$$

has a local minimum in $N$, see Figure 7 in the Appendix. Section 4 shows that this is observed in experiments and this minimum also motivates the divide and conquer strategy we use in the next section.

# 3 A geometrically greedy Algorithm

Algorithm 1 produces candidates for the reduced measure by random sampling and then accepts or rejects them via the characterization given in Theorem 3. We now optimize the first part of it, namely the selection of candidates, by exploiting better the geometry of cones.

**Motivation in two dimensions.** Having chosen $\mathbf{x}^\star$ points we know that we have found a solution if $C^-(\mathbf{x}^\star) \cap \mathbf{x} \neq \emptyset$. Hence, maximizing the volume of the cone increases the chance that this intersection is not empty. Indeed, when $n = 2$ it is easy to show the following result.

**Theorem 4.** *Let $\mathbf{x}$ be a set of $N \geq 3$ points in $\mathbb{R}^2$ and $x_1 \in \mathbf{x}$. Define $\mathbf{x}^\star = (x_1^\star, x_2^\star)$, where*

$$x_2^\star := \arg\max_{x \in \mathbf{x} \setminus \{x_1^\star\}} \left| \frac{\langle x_1^\star, x \rangle}{\|x_1^\star\| \|x\|} - 1 \right|. \tag{5}$$

*There exists a convex combination $\sum_{x \in \mathbf{x}} w_x x$ of $\mathbf{x}$ that equals $0$ if and only if $\mathbf{x} \cap C^-(\mathbf{x}^\star) \neq \emptyset$.*

Theorem 4 follows immediately from the cone geometry but it is instructive to spell it out since it motivates the case of general $n$.

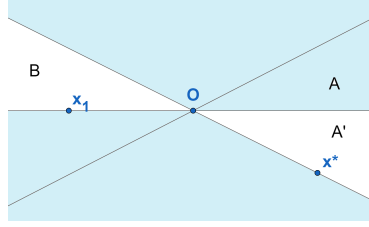

Figure 2: Proof of Theorem 4.

*Proof.* The shaded areas in Figure 2 indicate $C^-(x_1, x^\star)$ (on the top) and $C(x_1, x^\star)$ (on the bottom). Moreover, by definition of $x^\star$ in the region $A$ and $A'$ there are no points.

($\Rightarrow$) If there exists a point $x_2$ in $C^-(x_1, x^\star)$, then by convexity it follows that there exists a convex combination of 0 for $x_1, x_2, x^\star$.
($\Leftarrow$) If there does not exist a point in $C^-(x_1, x^\star)$, then there are points only in $B \cup C(x_1, x^\star)$, therefore, again for simple convex geometry arguments, it is impossible that there exists a convex combination of 0 for $x_1, x_2, x^\star$. □

Hence, if we modify Algorithm 1 by selecting in step 4 the new point by maximizing the angle according to (5) then for $n = 2$, Theorem 4 guarantees that $n + 1 = 3$ points out of $N$ are found that constitute a reduced measure $\hat{\mu}$ in $\tau \leq 2$ computational steps.

**A geometrically greedy Algorithm.** For general dimensions $n$, the intuition remains that a good (but not perfect) proxy to maximize the likelihood that the negative cone is non-empty, is given by maximizing the volume of the cone, see [14, Chapter 8]. Such a volume maximization is a long-standing open question and goes at least back to [15]; see [16] for an overview. One reviewer, also pointed to recent papers that implicitly apply a similar intuition to other problems, see [17, 18]. All this motivates the "*geometrically greedy*" Algorithm 2 that applies for any $n$. First note that the

---

**Algorithm 2** Optimized measure reduction algorithm

1: **procedure** REDUCE-OPTIMIZED(A set $\mathbf{x}$ of $N$ points in $\mathbb{R}^n$)
2:      Choose $n$ points $\mathbf{x}^\star = \{x_1^\star, \ldots, x_n^\star\}$ from $\mathbf{x}$
3:      $i \leftarrow 0$
4:      **while** $C^-(\mathbf{x}^\star) \cap \mathbf{x} = \emptyset$ **do**
5:          $\mathbf{x} \leftarrow \mathbf{x} \setminus \text{interior}\{C(\mathbf{x}^\star)\}$
6:          **if** $i = 0$ **then**
7:              $x_{i+1}^\star \leftarrow \arg\max_{x \in \mathbf{x} \setminus \mathbf{x}^\star} \left| \langle x, \sum_{j=2}^n x_j^\star \rangle - 1 \right|$
8:          **else**
9:              $x_{i+1}^\star \leftarrow \arg\max_{x \in \mathbf{x} \setminus \mathbf{x}^\star} \left| \langle x, \sum_{j=1}^i x_j^\star \rangle - 1 \right|$
10:         **end if**
11:         $i \leftarrow ((i+1) \mod n)$
12:      **end while**
13:      $\mathbf{x}^\star \leftarrow \mathbf{x}^\star \cup x^\star$ with an arbitrary $x^\star \in C^-(\mathbf{x}^\star) \cap \mathbf{x}$
14:      Solve the linear system $\sum_{x \in \mathbf{x}^\star} w_x^\star x = 0$ for $w^\star = (w_x^\star)_{x \in \mathbf{x}^\star}$
15:      **return** $(\mathbf{x}^\star, w^\star)$
16: **end procedure**

---

deletion of points in step 5 in Algorithm 2 does not throw away potential solutions: suppose we have deleted a point $\hat{x}$ in the previous step that was part of a solution, i.e. there exists a set of $n+1$ points $\hat{\mathbf{x}}^\star$ in $\mathbf{x}$ such that $\hat{x} \in \hat{\mathbf{x}}^\star$, and there exist $n+1$ weights $\hat{w}_i, i \in [0, \ldots, n+1], \hat{w}_i \in [0, 1], \sum w_i = 1$ such that $\sum_{i=1}^{n+1} \hat{w}_i \hat{x}_i^* = 0, \hat{x}_i^* \in \hat{\mathbf{x}}^\star$. If we indicate with $\mathbf{x}_c$ the $n$ vectors of the basis of the cone of the previous iteration, we know that $\hat{x} \in \text{interior}\{C(\mathbf{x}_c)\}$, which means there exist strictly positive values $c_i$ such that $\hat{x} = \sum_{i=1}^n c_i x_i^c, x_i^c \in \mathbf{x}^c$. Therefore,

$$\sum_{i=1}^{n+1} \hat{w}_i \hat{x}_i^* = \hat{w}_{n+1} \hat{x} + \sum_{i=1}^n \hat{w}_i \hat{x}_i^* = \hat{w}_{n+1} \sum_{i=1}^n c_i x_i^c + \sum_{i=1}^n \hat{w}_i \hat{x}_i^*, \; x_i^c \in \mathbf{x}^c \text{ and } \hat{x}_i^* \in \hat{\mathbf{x}}^\star.$$

Given that $w_i$ and $c_i$ are positive $0 \in \text{Conv}\{\hat{\mathbf{x}}^\star \cup \mathbf{x}_c\}$ and we can apply again the Caratheodory's Theorem 2, which implies that the deleted point $\hat{x}$ was not essential. The reason for the if clause in step 6 is simply that the first time the loop is entered we optimize using the randomly selected bases, but in subsequent runs it is intuitive that we should only optimize over the base points that were optimized in previous loops.

**Complexity.** We now discuss the complexity of Algorithm 2.

**Proposition 3.** *The complexity of Algorithm 2 to compute a reduced measure $\hat{\mu}$ ,as in Theorem 1, is*

$$O(n^3 + n^2 N) + (\tau - 1)O(n^2 + nN),$$

*here $\tau = \inf\{i \geq 1 : C^-(\mathbf{X}_i) \cap \mathbf{x} \neq \emptyset\}$ where $\mathbf{X}_1, \mathbf{X}_2, \ldots$ are obtained as in Algorithm 2.*

In contrast, to the complexity of Algorithm 1, Corollary 1, the $n^3$ term that results from a matrix inversion is no longer proportional to $\tau$, and the random runtime $\tau$ only affects the complexity proportional to $n^2 + nN$. For a generalization of Theorem 4 from $n = 2$ to general $n$, that is a statement of the form "in $n$ dimensions the algorithm terminates after at most $\tau \leq f(n)$", one ultimately runs into the result of a "positive basis" from discrete geometry, see for example [19], that says if $n \geq 3$ it is possible to build positive independent set of vectors of any cardinality. Characterizing the probability of the occurrences of such sets is an ongoing discrete geometry research topic and we have nothing new to contribute to this. However, despite the existence of such "positive independent sets" for $n \geq 3$, the experiments in Section 4 underlines the intuition that in the generic case, maximizing the angles is hugely beneficial also in higher dimension. If a deterministic bound on the runtime is crucial, one can combine the strengths of Algorithm 2 (a good chance of finding $\mathbf{x}^\star$ quickly by repeatedly smart guessing) with the strength of deterministic algorithms such as [4, 5, 6] by running Algorithm 2 for at most $k$ steps and if a solution is not found run a deterministic algorithms. Indeed, our experiments show that this is on average a very good trade-off since most of the probability mass of $\tau$ is concentrated at small $k$, see also Appendix E.

**Robustness.** An interesting question is how robust the measure reduction procedure is to the initial points. Therefore assume we know the solution of the recombination problem (RP) for $\mathbf{x} \subset \mathbb{R}^n$, i.e. a subset of $n + 1$ points $\hat{\mathbf{x}} = (\hat{x}_1, \hat{x}_2, \ldots, \hat{x}_{n+1}) \subset \mathbf{x}$ and a discrete measure $\hat{\mu}$ on $\hat{\mathbf{x}}$ such that $\hat{\mu}(\hat{\mathbf{x}}) = 0$. If a set of points $\mathbf{y}$ is close to $\mathbf{x}$ one would expect that one can use the solution of the RP for $\mathbf{x}$ to solve the RP for $\mathbf{y}$. The theorem below uses the Woodbury matrix identity to make this precise.

**Proposition 4.** *Assume that $\text{span}(\hat{\mathbf{x}}) = \text{span}(\hat{\mathbf{x}}_{-1}) = \mathbb{R}^n$, where $\hat{\mathbf{x}}_{-i} := \hat{\mathbf{x}} \setminus \hat{x}_i$. Denote with $\mathbf{X}$ a matrix which as has rows the vectors in $\mathbf{x}$. Suppose there exists an invertible matrix $R$ and another matrix $E$, such that $\mathbf{X} = \mathbf{Y}R + E$. Denote $\gamma_1 := (\hat{\mathbf{X}}_{-1}^\top)^{-1} \hat{X}_1^\top$, where $\hat{\mathbf{x}}$ is a solution to the RP $\mathbf{x}$. Assuming that the inverse matrices exist, $\hat{\mathbf{X}}R + E_{\hat{\mathbf{x}}}$ is a solution to the RP $\mathbf{y}$ if and only if*

$$\gamma_1^\top + E_{\hat{x}_1} R^{-1} A_1^\top \leq \left( \gamma_1^\top + E_{\hat{x}_1} R^{-1} A_1^\top \right) E_{\hat{\mathbf{x}}_{-1}} \left( I + R^{-1} A_1^\top E_{\hat{\mathbf{x}}_{-1}} \right) R^{-1} A_1^T$$

*where $E_y$ indicates the part of the matrix $E$ related to the set of vectors $y \subset \mathbf{x}$ and $A_1 = (\hat{\mathbf{X}}_{-1}^\top)^{-1}$.*

This is not only of theoretical interest, since the initial choice of a cone basis in Algorithm 2 can be decisive. For example, if we repeatedly solve a "similar" RP, e.g. $N$ points are sampled from the same distribution, then Proposition 4 suggests that after the first RP solution, we should use the points in the new set of points that are closest to the previous solution as initial choice of the cone basis.

**Las Vegas resets.** The only source of randomness in Algorithm 2 is the choice of $\mathbf{x}^\star$ in step 2. If this happens to be a bad choice, much computational time is wasted, or even worse, the Algorithm might not even terminate. However, like any other random "black box" algorithm one can stop Algorithm 2 if $\tau$ becomes large and then restart with a random basis $\mathbf{x}^\star$ sampled independently from the previous one. We call a sequence $\mathscr{S} = (t_1, t_2, \ldots)$ of integers a reset strategy where the $i$th entry $t_i$ denotes the number of iterations we allow after the $i$th time the algorithm was restarted, e.g. $\mathscr{S} = (10, 3, 6, \ldots)$ means that if a solution is not found after 10 loops, we stop and restart; then wait for at most 3 loop iterations before restarting; then 6, etc. Surprisingly, the question which strategy $\mathscr{S}$ minimises the expected run time of Algorithm 2 has an explicit answer. A direct consequence of the main result in [20] is that $\mathscr{S}^\star = c \times (1, 1, 2, 1, 1, 2, 4, 1, 1, 2, 1, 1, 2, 4, 8, 1 \ldots)$ achieves the best expected running time up to a logarithmic factor, i.e. by following $\mathscr{S}^\star$ the expected running time is $O(\mathbb{E}[\tau^\star] \log \mathbb{E}[\tau^\star])$

where $\tau^\star$ denotes the minimal expected run time under *any* reset strategy. Thus, although in general we do not know the optimal reset strategy $\tau^\star$ (which will be highly dependent on the points $\mathbf{x}$), we can still follow $\mathscr{S}^\star$ which guarantees that Algorithm 2 terminates and that its expected running time is within a logarithmic factor of the best reset strategy. Since Algorithm 2 uses $n$ updates of a cone basis, it is natural to take $c$ proportional to $n$ (in our experiments we fixed throughout $c = 2n$).

**Divide and conquer.** A strategy used in existing algorithms [4, 5, 6] is for a given $\mu = \sum_{i=1}^{N} w_i x_i$ to partition the $N$ points into $2(n+1)$ groups $I_1, \ldots, I_{2(n+1)}$ of approximately equal size, compute the barycenter $b_i$ for each group $I_i$, and then carry out any measure reduction algorithm to reduce the support of $\sum_{i=1}^{2(n+1)} \frac{w_i}{\sum_{j \in I_i} w_j} \delta_{b_i}$ to $n+1$ points. One can then repeat this procedure, see [4, Algorithm 2 on p1306] for details, and since each iteration halves the support, this terminates after $\log(N/n)$ iterations. Hence the total cost is $O(Nn + \log(N/n))C(2(n+1), n+1))$, where $C(2(n+1), n+1)$ denotes the cost to reduce the measure from $2(n+1)$ points to $n+1$ points. For the algorithm in [4] $C(2(n+1), n+1) = O(n^4)$, similarly to [6]; for the algorithm in [5] $C(2(n+1), n+1) = O(n^3)$. Similarly, we can also run our randomized Algorithm 2 on the $2(n+1)$ points. However, the situation is more subtle since the regime where Algorithm 2 excels is the big data regime $N \gg n$, so running the algorithm on smaller groups of points could reduce the effectiveness. Already for simple distributions and Algorithm 1, as in Proposition 2 item 2, we see that for the optimal choice we should divide the points in $N_n^*$ groups with $N_n^*$ denoting the argmin of the complexity in $N$. This decreases the computational cost to $O(Nn + \log_{N_n^*/n}(N)\bar{C}(N_n^*, n+1))$, where $\bar{C}$ denotes the computational cost of Algorithm 1 to reduce from $N_n^*$ points to $(n+1)$ points. In general, the optimal $N_n^*$ will depend on the distribution of the points, but an informal calculation shows that $N_n^* = 50(n+1)$ achieves the best trade-off; see Appendix D for details.

## 4 Experiments

We give two sets of experiments to demonstrate the properties of Algorithm 1 and Algorithm 2: (i) using synthetic data allows to study the behaviour in various regimes of $N$ and $n$, (ii) on real-world data, following [6], for fast least square solvers. As baselines we consider two recent algorithms [5] (*det3*) and [4] resp. [6] (*det4*)[1].

**Reducing empirical measures.** We sampled $N \in \{2n, 20, 30, 50, \ldots, 10^6\}$ points (i) a $n = 15$-dimensional standard normal random variable (*symmetric1*), (ii) a $n = 20$-dimensional standard normal random variable (*symmetric2*), (iii) $n = 20$ dimensional mixture of exponential (*non symmetric*). We then ran Algorithm 1 (*basic*), Algorithm 2 (*optimized*), as well as Algorithm 2 with the optimal Las Vegas reset (*optimized-reset*) and the divide and conquer strategy (*log-opt*); the results are shown in Figure 3. The first row clearly shows that the performance gets best in the big data regime $N \gg n$. The biggest effect is how the angle/volume optimization of Algorithm 2 drastically reduces the number of iterations compared to Algorithm 1, and therefore the running time and the variance. From a theoretical perspective is interesting that the shape predicted in Proposition 2, for symmetric distributions such as the normal distribution (the two columns on the left) also manifests itself for the non-symmetric mixture (the right column); see also Figure 7 in the Appendix. The Las Vegas reset strategy is only noticeable in the regime when simultaneously $N$ and $n$ are close and relatively large; e.g. Figure 3 falls not in this regime which is the reason why the plots are indistinguishable. Nevertheless, even in regimes such as in Figure 3 the reset strategy is at least on a theoretical level useful since it guarantees the convergence of Algorithm 2 by excluding pathological cases of cycling through a "sequence" of cone bases (although we have not witnessed such pathological cases in our experiments).

As expected, the regime when the number of points $N$ is much higher than the number of dimensions $n$, yields the best performance for the randomized algorithm; moreover, Figure 4 shows that the run time is approximately $O(n)$ (in contrast to the runtime of *det4* and *det3* that is $O(n^4)$ resp. $O(n^3)$).

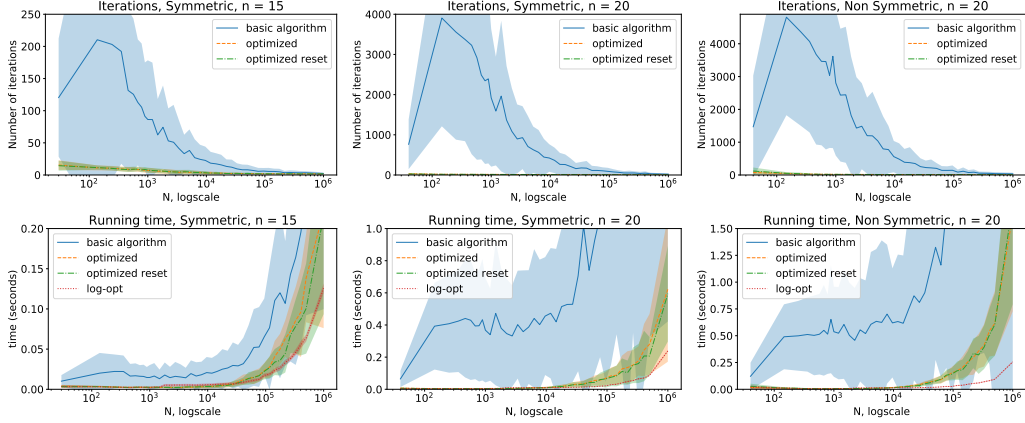

Figure 3: Running time and number of iterations of the randomized algorithms as *N* varies. The first two columns show the results for *symmetric1* and *symmetric2* , the right column for *non-symmetric*. The shaded area represents the standard deviation (from 70 repetitions of the experiment).

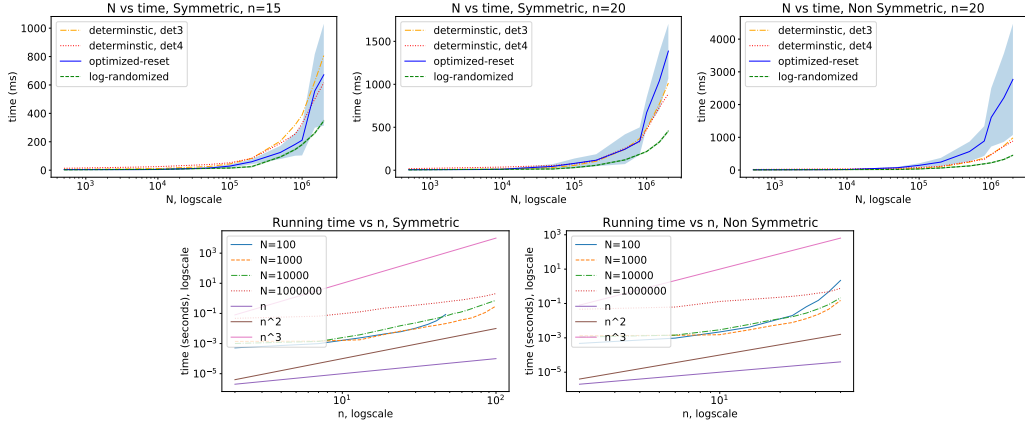

Figure 4: The top row compares the running time of randomized Algorithms against deterministic algorithms as *N* varies for *symmetric1* (left), *symmetric2* (middle) and *non-symmetric* (right). The shaded area represents the standard deviation (from 70 repetitions of the experiment). The bottom row running time of the *log-optimized* algorithm as *n* varies for different *N* (average of 70 samples).

**Fast mean-square solvers.** In [6] a measure reduction was used to accelerate the least squares method, i.e. the solution of the minimization problem $\min_w \|\mathbf{X}w - \mathbf{Y}\|^2$ where $\mathbf{X} \in \mathbb{R}^{N \times d}$ and $\mathbf{Y} \in \mathbb{R}^N$; as in Proposition 4, $\mathbf{X}$ denotes a matrix which has as row vectors the elements of $\mathbf{x}$, similarly for $\mathbf{Y}$. Sometimes precise solutions are required and in this case, the measure reduction approach yields a scalable method: Theorem 2 guarantees the existence of a subset of $n + 1 = (d+1)(d+2)/2 + 1$ points $(\mathbf{x}^\star, \mathbf{y}^\star)$ of $\mathbf{x}$ and $\mathbf{y}$ such that $(\mathbf{X}|\mathbf{Y})^\top \cdot (\mathbf{X}|\mathbf{Y}) = (\mathbf{X}^\star|\mathbf{Y}^\star)^\top \cdot (\mathbf{X}^\star|\mathbf{Y}^\star)$ where we denote with $(\mathbf{X}|\mathbf{Y})$ the element of $\mathbb{R}^{N \times (d+1)}$ formed by adding $\mathbf{Y}$ as a column. However, this implies that $\|\mathbf{X}w - \mathbf{Y}\|^2 = \|\mathbf{X}^\star w - \mathbf{Y}^\star\|^2$ for every $w$, hence it is sufficient to solve the least square problem in much lower dimensions once $\mathbf{X}^\star$ and $\mathbf{Y}^\star$ have been found. We use the following datasets from [6](i) 3D Road Network [21] that contains 434874 records and use the two attributes longitude and latitude to predict height, (ii) Household power consumption [22] that contains 2075259 records and use the two 2 attributes active and reactive power to predict voltage. We also add a synthetic dataset, (iii) where $\mathbf{X} \in \mathbb{R}^{N \times n}$, $\theta \in \mathbb{R}^n$ and $\varepsilon \in \mathbb{R}^N$ are normal random variables, and $\mathbf{Y} = \mathbf{X}\theta + \varepsilon$ which allows to study various regimes of *N* and *n*. Figure 5 shows the performance of Algorithm 2 with the Las Vegas reset, with the Las Vegas reset and the divide and conquer optimization on the datasets (i),(ii). We observe that already Algorithm 2 with Las Vegas resets is on average faster but the running time distribution has outliers where the algorithm takes longer than for the deterministic

run time algorithms; combined with divide and conquer the variance is reduced by a lot. Figure 6 shows the results on the synthetic dataset (iii) for various values of $N$ and $n = (d+1)(d+2)/2$.

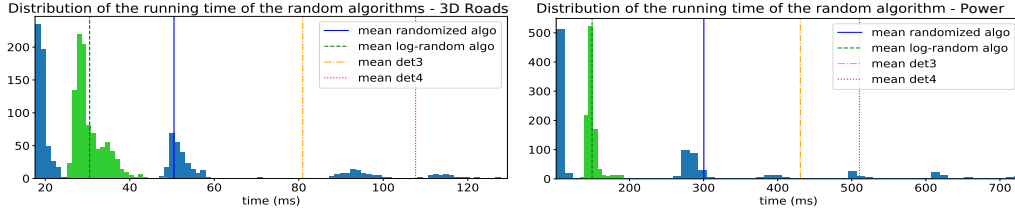

Figure 5: Histogram of the running time of Algorithm 2 with a reset strategy and of the "divide and conquer" variation algorithm (*log-random*). The vast majority of probability mass of the random runtime is below any of runtimes of the deterministic algorithms although with small probability it can take longer than the deterministic runtimes.

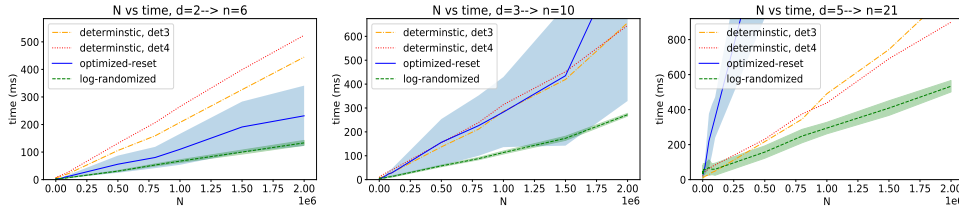

Figure 6: Performance of the different algorithms on the synthetic data set (iii) for various values of $N$ and $n = (d+1)(d+2)/2$.

**Breaking the randomized Algorithm.**   As the above experiments show, Algorithm 2 can lead to big speed ups. However, it is not uniformly better than the deterministic algorithms, and there are situations where one should not use it: firstly, Algorithm 2 was optimized to work well in the $N \gg n$ regime and while this is an important regime for data science, the recombination problem itself is of interest also in other regimes [4]. Secondly, the Las Vegas resets give a finite running time, but it is easy to construct examples where this can be much worse than the deterministic algorithms. Arguably the most practically relevant issue is when the independence hypothesis of Theorem 3 is not satisfied. This can appear in data sets with a high number of highly correlated categorical features, such as [23]. This can be overcome by using the Weyl Theorem, see Remark 1 in Appendix A but the computational cost is higher than computing the inverse of the cone basis ($A$ in Theorem 3) and the benefits would be marginal, if not annulled, compared to the deterministic algorithms. More relevant is that Algorithm 2 can be easily combined with any of the deterministic algorithms to build an algorithm that has a worst case run time of the same order as a deterministic one but has a good chance of being faster; see Appendix E for details and experiments.

## 5   Summary

We introduced a randomized algorithm that reduces the support of a discrete measure supported on $N$ atoms down to $n+1$ atoms while preserving the statistics as captured with $n$ functions. The key was a characterization of the barycenter in terms of negative cones, that inspired a greedy sampling. Motivated by the geometry of cones this greedy sampling can be optimized, and finally combined with optimization methods for randomized algorithms. This yields a "greedy geometric sampling" that follows a very different strategy than the previous deterministic algorithms, and that performs very well in the big data regime when $N \gg n$ as is often the case for large sample sizes common in inference tasks such as least square solvers.

## Broader Impact

The authors do not think this section is applicable to the present work, this work does not present any foreseeable societal consequence.

## Acknowledgements and Disclosure of Funding

The authors want to thank The Alan Turing Institute and the University of Oxford for the financial support given. FC is supported by The Alan Turing Institute, TU/C/000021, under the EPSRC Grant No. EP/N510129/1. HO is supported by the EPSRC grant "Datasig" [EP/S026347/1], The Alan Turing Institute, and the Oxford-Man Institute.

## Footnotes

[1]Although the derivation is different, the resulting algorithms in [4, 6] are essentially identical; we use the implementation of [6] since do the same least mean square experiment as in [6]. All experiments have been run on a MacBook Pro, CPU: i7-7920HQ, RAM: 16 GB, 2133 MHz LPDDR3.

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
