[Supplementary Material]

# A  Properties of Algorithm 1

**Background on polytopes.**  To prepare the proof of Theorem 3 we want to recall that a well-known tool from discrete geometry, is that polygons and polyhedral descriptions are equivalent. That is $C(\mathbf{x})$ is an affine span of the vectors $\mathbf{x}$ as in Definition 1, but equivalently $C(\mathbf{x})$ is the intersection of hyperplanes. In general this is the content of the celebrated Weyl–Minkowksi theorem and computing one representation from the other is non-trivial, see [23]. However, when restricted to $n$ generic vectors in $\mathbb{R}^n$ (as is the case required in Theorem 3), one can immediately switch from one to the other, see item 2 of Theorem 3.

**Proofs of Theorem 3 and Proposition 1**

**Theorem 3.** *Let* $\mathbf{x} = \{x_1, \ldots, x_{n+1}\}$ *be a set of* $n+1$ *points in* $\mathbb{R}^n$ *such that* $\mathbf{x} \setminus \{x_{n+1}\}$ *spans* $\mathbb{R}^n$. *Let* $A$ *be the matrix that transforms* $\mathbf{x} \setminus \{x_{n+1}\} = \{x_1, \ldots, x_n\}$ *to the orthonormal basis* $\{e_1, \ldots, e_n\}$ *of* $\mathbb{R}^n$, *i.e.* $Ax_i = e_i$. *Further, let* $h_i$ *be the unit vector such that* $\langle h_i, x \rangle = 0$ *for all* $x \in \mathbf{x} \setminus \{x_i, x_{n+1}\}$ *and* $\langle h_i, x_i \rangle < 0$ *and denote with* $H_{\mathbf{x} \setminus \{x_{n+1}\}}$ *a* $n \times n$ *matrix that has* $h_1, \ldots, h_n$ *as row vectors. It holds that*

1. $C(\mathbf{x} \setminus \{x_{n+1}\}) = \{c | H_{\mathbf{x} \setminus \{x_{n+1}\}} c \leq 0\}$ *and* $C^-(\mathbf{x} \setminus \{x_{n+1}\}) = \{c | H_{\mathbf{x} \setminus \{x_{n+1}\}} c \geq 0\}$.

2. $Ax \geq 0$ *if and only if* $H_{\mathbf{x} \setminus \{x_{n+1}\}} x \leq 0$ *and* $Ax \leq 0$ *if and only if* $H_{\mathbf{x} \setminus \{x_{n+1}\}} x \geq 0$.

3. *There exists a convex combination of* $\mathbf{x}$ *with* $0$ *as barycentre,* $\sum_{i=1}^{n+1} w_i x_i = 0$ *for some* $w_i > 0$, *and* $\sum_{i=1}^{n+1} w_i = 1$ *if and only if* $x_{n+1} \in C^-(\mathbf{x} \setminus \{x_{n+1}\})$.

*Proof.* For item 1 first note that the $h_i$ are well-defined since any set of $n-1$ independent points determines a hyperplane that includes $0$ and that divides $\mathbb{R}^n$ into two parts. Each of these two parts is of the form $\{x : \langle h, x \rangle \leq 0\}$ or $\{x : \langle h, x \rangle > 0\}$ and the additional condition $\langle h, x_i \rangle < 0$ selects one of the two parts. Now let $c = \sum_{x \in \mathbf{x} \setminus \{x_{n+1}\}} w_x x$ be a general vector. Since $H_{\mathbf{x} \setminus \{x_{n+1}\}} x \leq 0$, for $x \in \mathbf{x} \setminus \{x_{n+1}\}$, it follows that $H_{\mathbf{x} \setminus \{x_{n+1}\}} c = \sum_{x \in \mathbf{x} \setminus \{x_{n+1}\}} w_x H_{\mathbf{x} \setminus \{x_{n+1}\}} x$ satisfies $H_{\mathbf{x} \setminus \{x_{n+1}\}} c \leq 0$ if and only if the $w_x$, for $x \in \mathbf{x} \setminus \{x_{n+1}\}$, are positive.

For item 2, we can write $x = \sum_{i=1}^n w_i x_i$, for some $w_i \in \mathbb{R}$, hence $Ax = (w_1, \ldots, w_n)^\top$. By definition

$$H_{\mathbf{x} \setminus \{x_{n+1}\}} x = \sum w_i H_{\mathbf{x} \setminus \{x_{n+1}\}} x_i = \sum w_i (\langle h_1, x_i \rangle, \ldots, \langle h_n, x_i \rangle)^\top = (w_1 \langle h_1, x_1 \rangle, \ldots, w_n \langle h_n, x_n \rangle)^\top.$$

Note that $\langle h_i, x_i \rangle \leq 0$ by definition, therefore sign $\{-Ax\}$ = sign $\{H_{\mathbf{x} \setminus \{x_{n+1}\}} x\}$. The statement with the reversed inequalities follows similarly.

For item 3($\Rightarrow$) assume there exists a convex combination of $\mathbf{x}$, this means that $x_{n+1} = -\frac{1}{w_{n+1}} \sum_{i=1}^n w_i x_i$, and

$$Ax_{n+1} = -\frac{1}{w_{n+1}} \sum_{i=1}^n w_i Ax_i = \sum_{i=1}^n -\frac{w_i}{w_{n+1}} e_i.$$

Therefore, $Ax_{n+1} \leq 0$ which is by item 2 equivalent to $H_{\mathbf{x} \setminus \{x_{n+1}\}} x \geq 0$. Thus, $x_{n+1} \in C^-(\mathbf{x} \setminus \{x_{n+1}\})$. Finally, for item 3($\Leftarrow$) assume that $x_{n+1} \in C^-(\mathbf{x} \setminus \{x_{n+1}\})$. The by item 2 $Ax_{n+1} \leq 0$. Moreover, $\exists \lambda_i \in \mathbb{R}$ such that $x_{n+1} = \sum_{i=1}^n \lambda_i x_i$, therefore $Ax_{n+1} = \sum_{i=1}^n \lambda_i e_i = (\lambda_1, \ldots, \lambda_n)^\top \leq 0$. Let us call $\lambda^* := 1 - \sum_{i=1}^n \lambda_i$, by the decomposition of $x_{n+1}$ we know that

$$\frac{1}{\lambda^*} x_{n+1} + \sum_{i=1}^n \frac{-\lambda_i}{\lambda^*} x_i = 0 \text{ and } \frac{1}{\lambda^*} + \sum_{i=1}^n \frac{-\lambda_i}{\lambda^*} = 1 \text{ and } \frac{1}{\lambda^*}, \frac{-\lambda_i}{\lambda^*} \geq 0.$$

$\square$

**Remark 1.** *The assumption that* $\{\mathbf{x} \setminus \{x_{n+1}\}\}$ *spans* $\mathbb{R}^n$ *can be relaxed, indeed item 1 is a particular case of the Weyl's Theorem, which briefly does not require the independence of the cone basis. From an implementation point of view, however, item 2 gives an important boost, indeed the computation of* $H_{\mathbf{x} \setminus \{x_{n+1}\}}$ *is heavier than inverting a matrix, i.e. computing* $A$, *since it requires the computation of the coefficients of* $n$ *different hyperplanes in* $\mathbb{R}^n$. *Moreover, speaking about the greedy searching strategy of Algorithm 2, using* $H_{\mathbf{x} \setminus \{x_{n+1}\}}$ *in place of* $A$ *does not allow the use of the Sherman–Morrison formula weighing even more on the total computational cost.*

**Proposition 1.** *Let $N > n+1$ and $\mu$ be a discrete probability with finite support and $f_1, \ldots, f_n$ be as in Theorem 1. Moreover wlog assume $\mathbb{E}_{X \sim \mu}[f_i(X)] = 0$ for $i = 1, \ldots, n$. With $p := \frac{n \cdot n!(N-n)!}{N!}$ it holds that $\mathbb{E}[\tau] \leq \frac{1}{p}$ and $Var(\tau) \leq \frac{1-p}{p^2}$ and, for fixed $n$, $\lim_{N \to \infty} \mathbb{E}[\tau] = 1$.*

*Proof.* Note that in every run through the loop, $n$ points are randomly selected. However, by Theorem 3, Algorithm 1 finishes when the event

$$A := \{\text{for } n \text{ uniformly at random chosen points } \mathbf{x}^\star \text{ from } \mathbf{x}, \exists x \in \mathbf{x} \; s.t. \; x \in C^-(\mathbf{x}^\star)\}.$$

occurs. Combined with Tchakaloff's Theorem guarantees this shows that there exists at least one set of $n$ points $\mathbf{x}^\star$ such that $C^-(\mathbf{x}^\star) \neq \emptyset$ and therefore

$$\mathbb{P}(A) \geq \frac{\binom{n+1}{n}}{\binom{N}{n}} = \frac{n \cdot n!(N-n)!}{N!},$$

By independence, $\tau$ can be modelled by a geometric distribution with parameter $p = \mathbb{P}(A)$

$$\mathbb{P}(\tau = k) = (1-p)^{k-1} p$$

and the bounds for mean and variance follow.

$\square$

# B  Properties when applied to special measures

**Proof of Proposition 2**

**Proposition 2.** *Let $N > n+1$ and let $f_1, \ldots, f_n$ be $n$ real-valued functions and $X_1, \ldots, X_N$ be $N$ i.i.d. copies of a random variable $X$. Set $F(X) = (f_1(X), \ldots, f_n(X))$, assume $\mathbb{E}[F(X)] = 0$ and denote*

$$E := \{0 \in \mathrm{Conv}\{F(X_i), i \in \{1, \ldots, N\}\}\}. \tag{2}$$

*1. $\mathbb{E}[\tau|E] \leq \frac{1}{p}$ and $Var(\tau|E) \leq \frac{1-p}{p^2}$, where*

$$p = \max\left\{\frac{n \cdot n!(N-n)!}{N!}, 1 - \mathbb{P}\left(0 \notin \mathrm{Conv}\{F(X_1), \ldots, F(X_{n+1})\}\right)^{N-n}\right\}, \tag{3}$$

*2. If the law of $F(X)$ is invariant under reflection in the origin, then $\mathbb{P}\left(0 \notin \mathrm{Conv}\{F(X_1), \ldots, F(X_{n+1})\}\right) = 1 - 2^{-n}$,*

*3. For fixed $n$, as $N \to \infty$*

$$\mathbb{P}(\text{for } n \text{ uniformly at random chosen points } \mathbf{x}^\star \text{ from } \mathbf{x}, \exists x \in \mathbf{x} \; s.t. \; x \in C^-(\mathbf{x}^\star)) \to 1,$$

*where $\mathbf{x} = \{F(X_1), F(X_2), \ldots, F(X_N)\}$.*

*Proof.* As in Proposition 1, the algorithm terminates when the event

$$A := \{\text{for } n \text{ uniformly at random chosen points } \mathbf{x}^\star \text{ from } \mathbf{x}, \exists x \in \mathbf{x} \; s.t. \; x \in C^-(\mathbf{x}^\star)\}$$

happens, where $\mathbf{x} = \{F(X_1), F(X_2), \ldots, F(X_N)\}$.
For item 1 denote $F_i := (f_1(X_{I_i}), \ldots, f_n(X_{I_i}))$ where $\{I_1, \ldots, I_N\}$ is a uniform shuffle of $\{1, \ldots, N\}$, i.e. a random permutation of its elements that makes every rearrangement equally probable, then

$$A = \{\exists i \in \{n+1, \ldots, N\} \; s.t. \; F_i \in C^-(F_1, \ldots, F_n)\}$$

and note that

$$\mathbb{P}(A|E) = \frac{\mathbb{P}(E|A)\mathbb{P}(A)}{\mathbb{P}(E)} \geq \mathbb{P}(A)$$

since by Theorem 3 $\mathbb{P}(E|A) = 1$ and $\mathbb{P}(E) > 0$ since $N \geq n+1$ and $\mathbb{E}F(X) = 0$. The estimate of Proposition 1 $\mathbb{P}(E|0 \in \text{Conv}\{F_i\}) \geq \frac{n \cdot n!(N-n)!}{N!}$ is still valid, moreover

$$\mathbb{P}(A|E) \geq \mathbb{P}(A) = \mathbb{P}(\exists i \in \{n+1,\ldots,N\} \text{ such that } F_i \in C^-(F_1,\ldots,F_n))$$

$$= 1 - \prod_{j=n+1}^{N} \mathbb{P}(F_j \notin C^-(F_1,\ldots,F_n))$$

$$= 1 - \mathbb{P}(F_{n+1} \notin C^-(F_1,\ldots,F_n))^{N-n}$$

$$= 1 - \mathbb{P}(0 \notin \text{Conv}(F_1,\ldots,F_{n+1}))^{N-n}$$

where the last equality follows from Theorem 3. We have therefore two different bounds for $\mathbb{P}(A|E)$, so we can take the maximum, i.e.

$$\mathbb{P}(A|E) \geq \max\left\{ \frac{n \cdot n!(N-n)!}{N!}, 1 - \mathbb{P}(0 \notin \text{Conv}\{F_1,\ldots,F_{n+1}\})^{N-n} \right\}$$

Item 2. In [24] the author shows that when the $F_i$ are distributed uniformly randomly on the unit sphere, then $\mathbb{P}(0 \notin \text{Conv}(F_1,\ldots,F_{n+1}))^{N-n} = (1-2^{-n})^{N-n}$. In [13][Theorem 8.2.1] it is shown the same result, for all the symmetric distributions with respect to 0.

Now $\tau$ can be modelled by a geometric distribution with parameter $p = \mathbb{P}(A|E)$, i.e. $\mathbb{P}(\tau = i) \geq (1-p)^{i-1}p$ and the mean and variance follows.

For item 3, it is enough to show $\mathbb{P}(A) \to 1$:

$$\mathbb{P}(A) = \mathbb{P}(A|E) \times \mathbb{P}(E) + \mathbb{P}\left(A|E^C\right) \times \mathbb{P}\left(E^C\right) = \mathbb{P}(A|E) \times \mathbb{P}(E) + 0 \times \mathbb{P}\left(E^C\right) \to 1 \times 1,$$

as $N \to \infty$, where $\mathbb{P}\left(A|E^C\right) = 0$ is due to Theorem 3, the convergence $\mathbb{P}(E) \to 1$ is guaranteed by Theorem 5 and the convergence $\mathbb{P}(A|E) \to 1$, is guaranteed by the proof of item 1 for fixed $n$. □

As intuition suggest, the event that the mean is included in the convex hull occurs almost surely.

**Theorem 5** ([8]). *Let $X_1,\ldots,X_N$ be i.i.d. samples from a random variable $X$ that has a first moment $\mathbb{E}[X] < \infty$. Then $\mathbb{P}(\mathbb{E}[X] \in \text{Conv}\{X_i\}_{i=1}^{N}) \to 1$, as $N \to \infty$.*

**Sampling from empirical measures**    Often we do not know the distribution $\phi$ of the points or $\mathbb{E}[F(X)]$, moreover it could be that for the realized samples $\{F_i\}_{i=1}^{N}$, $\mathbb{E}[F(X)] \notin \text{Conv}\{F_i\}_{i=1}^{N}$. In these cases, due to Theorem 2, and since Algorithm 1 is based on Theorem 3, which assumes that the barycentre of the points given is 0, the input of the Algorithm is not the collection $\{F_i\}_{i=1}^{N}$, but

$$\hat{F}_i = (f_1(X_i),\ldots,f_n(X_i)) - \left( \sum_{j=1}^{N} f_1(X_j)w_j,\ldots,\sum_{j=1}^{N} f_n(X_j)w_j \right),$$

in this way we are sure that the barycentre is 0 and $0 \in \text{Conv}\{\hat{F}_i\}_{i=1}^{N}$, and the hypothesis of both Theorem 2 and 3 are satisfied. Unfortunately the $\{\hat{F}_i\}_{i=1}^{N}$ are not independent, which leads to an impossible analysis, even though the correlation between the $\hat{F}_i$ decreases when $N$ becomes bigger and tends to 0. Thus, we can believe that the analysis of the proof of Proposition 2 is a good approximation of the complexity of the Algorithm 1, when the $\{\hat{F}_i\}_{i=1}^{N}$ are given as input and $N$ is big "enough", as it is shown in Figure 7 and Figure 3 in case of symmetric distribution.

It is relevant to note at this point that we can always consider uniform measures, i.e. $\mu = \frac{1}{N}\sum_{i=1}^{N} \delta_{x_i}$, modifying the support of the measure, and then eventually go back to the original (not-uniform) measure.

**Lemma 1.** *Let us consider a set $\mathbf{x} = \{x_i\}_{i=1}^{N}$ in $\mathbb{R}^n$ and a sequence $\{\kappa_i\}_{i=1}^{N}$ of strictly positive numbers. There exists a measure $\mu$ on $\mathbf{x}$ such that $\mu(\mathbf{x}) = 0$ if and only if there exists a measure $\mu^\star$ on $\{\frac{x_i}{\kappa_i}\}_{i=1}^{N}$ such that $\mu^*(\{\frac{x_i}{\kappa_i}\}_{i=1}^{N}) = 0$.*

*Proof.* Let us assume that there exists $\mu$ on $\mathbf{x}$ such that $\mu(\mathbf{x}) = 0$, and let us call $\mu_i := \mu(x_i)$. It is enough to define $\mu^\star = \mu_i\kappa_i$. The other side of the equivalence is proved in the same way. □

**Remark 2.** *Lemma 1 is a consequence of the fact that a cone is defined only by the directions of the vectors of the "basis", and not from their length.*

Proposition 2 shows us a "universal strategy" to explore the space of all the combination of points more efficiently, i.e. choosing the basis of the cone to maximize the probability placed on its inverse. In other words, ideally we should try to maximize

$$\max_{F_i \in \mathbf{x}} \mathbb{P}\left(F(X) \in C^-(F_1, \ldots, F_n)\right).$$

Figure 7: The plots shows the logplot of Equation (4). It can be seen that this is the same shape obtained with the experimental simulations in Section 4.

Figure 7 shows the complexity of Algorithm 1 in case of symmetric distributions; it can be noticed that it has a local minima $N_n^*$.

## C   Properties of Algorithm 2

**Complexity of Algorithm 2**

**Proposition 3.** *The complexity of Algorithm 2 to compute a reduced measure $\hat{\mu}$, as in Theorem 1, is*

$$O(n^3 + n^2 N) + (\tau - 1)O(n^2 + nN),$$

*here $\tau = \inf\{i \geq 1 : C^-(\mathbf{X}_i) \cap \mathbf{x} \neq \emptyset\}$ where $\mathbf{X}_1, \mathbf{X}_2, \ldots$ are obtained as in Algorithm 2.*

*Proof.* The most expensive steps are the same ones as in Algorithm 1 and in addition the maximization problem. After the first step, given that we update one point of the basis at time, we can be more efficient using the Sherman–Morrison formula. Let us call $\mathbf{X}_t^\star$ the matrix whose rows are the vectors of the basis at the step $t$ $\mathbf{x}_t^\star$, therefore we have $A_t = ((\mathbf{X}_t^\star)^\top)^{-1}$, thus (shifting properly the vectors)

$$A_{t+1} = \left((\mathbf{X}_t^\star)^\top + (X^\star - X_1^\star)\cdot e_1^\top\right)^{-1} = A_t - \frac{A_t(X^\star - X_1^\star)e_1^\top A_t}{1 + e_1^\top A_t(X^\star - X_1^\star)},$$

in the case we want to substitute the "first" vector of the basis $X_1^\star$ with the vector $X^\star$. Let us note that the only multiplications to be computed are $A_t(X^\star - X_1^\star)$ and $[A_t(X^\star - X_1^\star)] \cdot [e_1^\top A_t]$, which are done in $O(n^2)$ operations. To check if there are points inside the cone (or the inverse cone), we multiply the matrix $A$ times the matrix $\mathbf{X}$ (of all the remaining vectors $\mathbf{X}$), and again after the first step costs $O(Nn^2)$, we can use the Sherman–Morrison formula as before and obtain

$$A_{t+1}\mathbf{X}^\top = A_t\mathbf{X}^\top - \frac{A_t(X^\star - X_1^\star)e_1^\top A_t\mathbf{X}^\top}{1 + e_1^\top A_t(X^\star - X_1^\star)}.$$

Let us note that we have already computed $A_t\mathbf{X}^\top$ at the previous step, $A_t(X^\star - X_1^\star)$ to compute $A_{t+1}$, therefore the only cost is to compute $[A_t(X^\star - X_1^\star)] \cdot [e_1^\top A_t\mathbf{X}^\top]$, which is done in $O(nN)$ operations. The previous computations show us that after the first step, updating one element at a time, improves the computational efficiency of the successive steps of a factor $n$. Let us now tackle the maximization problem, it requires to compute the norm, i.e. $O(Nn^2)$ operations, which could be done only once. Moreover, the maximization problem requires to compute (part of) the sum of the vectors in $\mathbf{x}^\star$ and then the scalar product, which require $O(Nn)$. The last expensive operation we should consider is due to solve the last system to find the weights. The system we want to solve is

$$\begin{pmatrix} (\mathbf{X}^\star)^\top & X^\star \\ \mathbf{1} & 1 \end{pmatrix} w = \begin{pmatrix} \mathbf{0} \\ 1 \end{pmatrix},$$

where $\mathbf{0}$ is a $n \times 1$ vector of 0, and $\mathbf{1}$ is a $1 \times n$ vector of 1. Using again the Sherman–Morrison formula, since we have already computed $A = (\mathbf{X}^\star)^{-1}$ the weights $w_i$ can be computed as

$$
w = \begin{pmatrix} (\mathbf{X}^\star)^\top & X^\star \\ \mathbf{1} & 1 \end{pmatrix}^{-1} \begin{pmatrix} \mathbf{0} \\ 1 \end{pmatrix} = \begin{pmatrix} A^\top + A^\top X^\star c^{-1} \mathbf{1} A^\top, & -A^\top X^\star c^{-1} \\ -c^{-1} \mathbf{1} A^\top, & c^{-1} \end{pmatrix} \begin{pmatrix} \mathbf{0} \\ 1 \end{pmatrix}
$$
$$
= -\frac{1}{c} \begin{pmatrix} A^\top X^\star \\ 1 \end{pmatrix},
$$

where $c = 1 - \mathbf{1} A^\top X^\star$ is a number. In this way we need $O(n^2)$ operations, not $O(n^3)$, i.e. the complexity of solving a linear system.
The total cost therefore is $O(n^3 + n^2 N) + (\kappa - 1) O(n^2 + nN)$. $\qquad\square$

**Remark 3.** *The gain in the computational cost we obtain using the Sherman–Morrison formula has a cost in term of numerical stability.*

### Robustness of the solution.

**Proposition 4.** *Assume that* $\mathrm{span}(\hat{\mathbf{x}}) = \mathrm{span}(\hat{\mathbf{x}}_{-1}) = \mathbb{R}^n$*, where* $\hat{\mathbf{x}}_{-i} := \hat{\mathbf{x}} \setminus \hat{x}_i$*. Denote with* $\mathbf{X}$ *a matrix which as has rows the vectors in* $\mathbf{x}$*. Suppose there exists an invertible matrix R and another matrix E, such that* $\mathbf{X} = \mathbf{Y} R + E$*. Denote* $\gamma_1 := (\hat{\mathbf{X}}_{-1}^\top)^{-1} \hat{X}_1^\top$*, where* $\hat{\mathbf{x}}$ *is a solution to the RP* $\mathbf{x}$*. Assuming that the inverse matrices exist,* $\hat{\mathbf{X}} R + E_{\hat{\mathbf{x}}}$ *is a solution to the RP* $\mathbf{y}$ *if and only if*

$$
\gamma_1^\top + E_{\hat{x}_1} R^{-1} A_1^\top \le \left( \gamma_1^\top + E_{\hat{x}_1} R^{-1} A_1^\top \right) E_{\hat{\mathbf{x}}_{-1}} \left( I + R^{-1} A_1^\top E_{\hat{\mathbf{x}}_{-1}} \right) R^{-1} A_1^T
$$

*where* $E_y$ *indicates the part of the matrix E related to the set of vectors* $y \subset \mathbf{x}$ *and* $A_1 = (\hat{\mathbf{X}}_{-1}^\top)^{-1}$*.*

*Proof.* From Theorem 3, we know that $\hat{\mathbf{X}} R + E_{\hat{\mathbf{x}}}$ is a solution if and only if $\left( \left( \hat{\mathbf{X}}_{-1} R + E_{\hat{\mathbf{x}}_{-1}} \right)^\top \right)^{-1} \left( \hat{X}_1 R + E_{\hat{x}_1} \right)^\top \le 0$. Let us note that the last product is a vector, therefore we can study the transpose and using the Woodbury matrix identity we have that

$$
\left( \hat{X}_1 R + E_{\hat{x}_1} \right) \left( \hat{\mathbf{X}}_{-1} R + E_{\hat{\mathbf{x}}_{-1}} \right)^{-1} = \left( \hat{X}_1 + E_{\hat{x}_1} R^{-1} \right) \left( \hat{\mathbf{X}}_{-1} + E_{\hat{\mathbf{x}}_{-1}} R^{-1} \right)^{-1}
$$
$$
= \left( \hat{X}_1 + E_{\hat{x}_1} R^{-1} \right) \left( I - A_1^T E_{\hat{\mathbf{x}}_{-1}} \left( I + R^{-1} A_1^T E_{\hat{\mathbf{x}}_{-1}} \right) R^{-1} \right) A_1^T.
$$

Setting the last equation less or equal than 0 shows the result. $\qquad\square$

This also implies that the solution is invariant under rotations.

## D Divide and conquer, choice of the subgroup size

As mentioned in Section 3, to apply a divide and conquer strategy requires to balance the size of subgroups against the property of Algorithm 2 to exploit a large number of points as to maximize the likelihood of points being in the (inverse) cone. Let us explain how we have chosen $N_n^* = 50(n+1)$, which should be thought as linear approximation of the exact minimum for the complexity of Algorithm 2. Therefore first note, that as Figure 8 shows, choosing any number between 20 and 80,

Figure 8: Bottom-right: it is shown how much the *optimized-reset* algorithm is faster than *det4* in case $N = (n+1) \times$ factor.

in place of 50, has similar effects if $n < 70$ in the case of symmetric distribution, whilst the same

holds in the case of mixture of exponentials (non symmetric) if $n \leq 40$. We think that this effect is due to the fact that "experimentally" there exists a long plateau in the running time of the optimized Algorithms with and without reset, see Figure 3. Therefore, let us suppose that there exist $k, K$ s.t. for any $k(n+1) \leq N^{(1)}, N^{(2)} \leq K(n+1)$ and $n < 40$, then $\bar{C}(N^{(1)}, n+1) \approx \bar{C}(N^{(2)}, n+1)$, where $\bar{C}(\cdot, n+1)$ is the computational cost to reduce $\cdot$ number of points in $\mathbb{R}^n$ using Algorithm 2. Moreover, we suppose that the $\arg\min_x \bar{C}(x, n+1) \in [k(n+1), K(n+1)]$. The previous two conditions are equivalent to the presence of the plateau in Figure 3, in correspondence with the minimum value of the running time, for the optimized Algorithms with and without reset. Under these assumptions, the best choice would be $N_n^* = K(n+1)$. Without knowing the value of $K$, however we can estimate the difference into the complexity for different group subdivisions: if we have $N \gg K(n+1)$ number of points, using the "divide and conquer" paradigm with $N^{(i)}$ groups, we can build two algorithms s.t. the difference of the computational costs is

$$O\left(Nn + \log_{N^{(1)}/n}(N/n)C(N^{(1)}, n+1)\right) - O\left(Nn + \log_{N^{(2)}/n}(N/n)C(N^{(2)}, n+1)\right) \approx \quad (6)$$

$$\approx \bar{C}(Kn, n+1)\log_{N^{(2)}/n}(N/n)\left(\frac{1}{\log_{N^{(2)}/n}N^{(1)}/n} - 1\right).$$

Therefore, we have that the difference depends on a factor $|1/\log_{N^{(2)}/n}(N^{(1)}/n) - 1|$. Given Figure 8, we have estimated approximately $k = 20$, $K = 80$ for the symmetric case, thus as a rule of thumb we assume that $N_n^* = 50(n+1)$ is a reasonable value, and in view of Equation (6) we can say that changing slightly 50 the running time would remain stable. The analogous argument can be made for the mixture of exponentials.

## E  A hybrid algorithm.

As mentioned in the introduction, the strategy of our randomized Algorithm 2 is very different to the deterministic ones, and one can combine both to form a new algorithm. The randomized Algorithm 2 runs into trouble when the independence assumption for the cone basis in Theorem 3 is not met which can happen in datasets with highly correlated features; on the other hand, the deterministic algorithms have the disadvantage that they need to complete a full run over the whole dataset even when geometric greedy sampling could have finished much earlier. We give the details for this hybrid Algorithm 3 below; it has a worst case running time of the same order as the deterministic Algorithms [4, 5, 6] but in return has a very good chance of terminating faster. We demonstrate this by benchmarking it against the same datasets for fast least square solvers that were used in [6].

Figure 9: The running time of the Combined Algorithm 3. Note that the mean of the log-random Algorithm is essentially equal to the one of the Combined Algorithm, except for the dataset [22] where the random-algo does not work (cf. discussion above). For [22] the features are highly correlated, so Algorithm 2 fails, but the additional time necessary to Algorithm 3 is only due to checking if the sampled basis are invertible; hence, in this case Algorithm 3 behaves as it was deterministic. We have chosen #_trials= 10 and $G = 50(n+1)$.

Some observations: (i) Algorithm 3 always finds a solution; (ii) in the case of dataset "with linear dependence", e.g. [22], if PCA reductions are not allowed as in the case of the application of [6], the basis **b** won't be invertible and therefore the complexity of the Algorithms in [5, 4, 6] would worsen of "only" the complexity to check #_trials times if a $n \times n$ matrix is invertible, i.e. #_trials$\times O(n^3)$, see Figure 9; (iii) Algorithm 2 must be run without reset, however note that we can add a reset strategy changing the number of iterations allowed to Algorithm 2 at step 8; (iv) following the guidelines of Section D, $G = 50(n+1)$ for "small $n$".

---

**Algorithm 3** Combined measure reduction algorithm

---
 1: **procedure** REDUCE-COMBINED(A set $\mathbf{x}$ of $N$ points in $\mathbb{R}^n$, $\mu = \{w_i\}$)
 2:     rem_points $\leftarrow N$
 3:     **while** rem_points $> n + 1$ **do**
 4:         Subdivide the points $\mathbf{x}$ in $G \wedge$ rem_points groups $\{\mathbf{x}_j\}_{j=1}^{G \wedge \text{rem\_points}}$
 5:         Compute $\bar{\mathbf{w}}_j = \sum_{w_i : x_i \in \mathbf{x}_j} w_i$, $\bar{\mathbf{x}}_j = \sum_{x_i \in g_j} w_i x_i / \bar{\mathbf{w}}_j$
 6:         **for** #_trials times **do**
 7:             $\mathbf{b} \leftarrow n$ random vectors from $\{\bar{\mathbf{x}}_j - \sum_j \bar{\mathbf{w}}_j \mathbf{x}_j\}_{j=1}^G$
 8:             $\mathbf{x}^\star, w^\star \leftarrow$ Algorithm 2 with the points $\{\bar{\mathbf{x}}_j - \sum_j \bar{\mathbf{w}}_j \mathbf{x}_j\}$ using $\mathbf{b}$ as cone basis
 9:             **if** Algorithm 2 has found a solution **then**
10:                 Exit for
11:             **end if**
12:         **end for**
13:         **if** Algorithm 2 has **not** found a solution **then**
14:             $\mathbf{x}^\star, w^\star \leftarrow$ Deterministic Algorithm (e.g. [5, 4, 6]) with measure $\{\bar{\mathbf{x}}_j\}$, $\{\bar{\mathbf{w}}_j\}$
15:         **end if**
16:         $\mathbf{x} \leftarrow \mathbf{x} \setminus \{x_i \text{ s.t. } x_i \in \mathbf{x}_j \text{ and } \bar{\mathbf{x}}_j \in \mathbf{x}^\star\}$                    ▷ Eliminate the points
17:         rem_points $\leftarrow N - \text{Cardinality}(\{x_i \text{ s.t. } x_i \in \mathbf{x}_j \text{ and } \bar{\mathbf{x}}_j \in \mathbf{x}^\star\})$
18:         $\{w_i\} \leftarrow \{w_i\} \setminus \{w_i \text{ s.t. } x_i \in \mathbf{x}_j \text{ and } \bar{\mathbf{x}}_j \in \mathbf{x}^\star\}$
19:         $\{w_i\} \leftarrow \{w_i \times w_j^\star \text{ s.t. } x_i \in \bar{\mathbf{x}}_j\}$                    ▷ Recalibrate the weights
20:     **end while**
21:     **return** $(\mathbf{x}^\star, w^\star)$
22: **end procedure**

---

To sum up, as Figure 9[2], the hybrid Algorithm 3 takes advantage of both the greedy geometric sampling of Algorithm 2 and robustness of the deterministic Algorithms for highly correlated datasets. Totalled over all the datasets Algorithm 3 is the fastest. Many variations of the above hybrid algorithm are of course possible.

## F   Implementation and benchmarking.

For *det4* we have used the code provided by the authors of [6] available at the repository[3] in Python; for *det3* the code of [5] has not been written in Python, therefore we have implemented it to allow for a fair comparison using standard Numpy libraries. We used throughout the same codeblocks for the Divide and Conquer strategy in all the implementations. We did not use the tree data-structure in [5] since it does not not change complexity bounds and is independent of the reduction procedure itself; however, it could be also used for *det4* and our randomized algorithms. Code for all experiments is available in public repository[4].

## Footnotes

[2]For practical reasons in our implementation of Algorithm 3 we have used our implementation of *det3*, see Section F.

[3]`https://github.com/ibramjub/Fast-and-Accurate-Least-Mean-Squares-Solvers`

[4]`https://github.com/FraCose/Recombination_Random_Algos`