[Reviews · NeurIPS 2020]

Review 1

Summary and Contributions: This paper tackles the problem of reducing the support of a measure, while preserving the expectation (w.r.t the above measure) of n given functions.

Strengths: The authors propose a (uniform) random sampling based algorithm as well as a more greedy approach for this problem. The algorithm

Weaknesses: While the algorithm seems to perform well empirically, the theoretical guarantees on the runtime are not very encouraging. How do these theoretical bounds compare with the bounds (if any) that are available for the algorithms in existing literature e.g. [4,5,6]? The bounds for the geometric greedy algorithm are not substantially better-- I was hoping to see that there would be a decrease in the number of iterations needed.

Correctness: I have not checked the supplementary. The approach seems reasonable, the empirical methodology is fine.

Clarity: Comments on the exposition given below.

Relation to Prior Work: I would like a better discussion of how these techniques relate to prior work. Please see below.

Reproducibility: Yes

Additional Feedback: I think the writing could be a little more expository. While I could follow the individual mathematical steps, the overall intuition behind the reduction is not clear to me, it would be nice if this can be brought out. For example, it might be nice, if the authors could add some pictures in the appendix to indicate what the negative cone really means geometrically. In effect it seems to be that the choice of n-points is random, modulo the fact that an n+1-th point can be found in the intersection of the negative cone and the set \bf{x}. Overall, given the runtimes presented in the paper, the practical significance of the result is not clear to me. In the least square regression example given, if we are fine with an approximation, there are results known that need only O(d*log(d) / eps^2), where (1+eps) = the approximation factor. To be fair, this technique gives an exact solution. But is it clear what benefits we hope to get by getting an exact instead of a close approximation? A related question that I had is the following-- in the general statement of what the authors are trying to prove, if we are with an approximation, is it possible to get a substantially more efficient algorithm? The equation between lines 141 and 142 has a typo, w_1 should be w_{n+1}. While the contribution seems quite nice, I would like to read a better explanation of what the benefit is over the approaches in [4, 5, 6]. It would also be nice to elaborate what the authors mean by saying that this approach is “complementary to previous work”.


Review 2

Summary and Contributions: This work addresses the problem of finding a discrete measure, which is supported on a subset of all data points, such that its expectation is equal to the original measure over the entire set. Authors propose a "greedy geometric sampling" heuristic to obtain the measure efficiently when the dimension is much smaller than the total number of points. This is validated in their experiments where they can combine their approach with existing methods to further reduce the runtime of a canonical least-squares problem.

Strengths: Concepts from cone geometry are used in the algorithm, which is novel. The experiments show that the reduction in computation time can be significant.

Weaknesses: The randomized algorithm can take a large value of resampling iterations, however authors give a "hybrid" algorithm that switches execution to existing deterministic algorithms when the run time exceeds a threshold. In the empirical evaluation, the advantage of resetting the algorithm over just running Algorithm 2 is not clear (both green and orange curves in Fig 1 look very similar).

Correctness: The method of algorithm 2 seems intuitive and I have checked its complexity statement (Proposition 3).

Clarity: The paper is written ok, though it would be better if some of the proofs could brought into the main text (for eg, Theorem 3, 4). Some typos: *line 125, should say "C^-(x*) \cap x \neq \empty" *display after line 141, should say "\hat{w}_{n+1}" after second equality

Relation to Prior Work: The references are from a wide variety of areas, as the authors say at line 31--32. The authors mention how their approach differs from previous work, but the core idea seems intuitive, and I wonder if it has been employed in other problem areas (some mentioned below)

Reproducibility: Yes

Additional Feedback: *The aim in algorithm 2 seems to be to select points that are as "spread out" as possible. Can that be encoded as an explicit optimization objective, and some "sparse greedy approximation" method [A] be employed to optimize it? *Another paper where vectors are selected to be as spread out is [B]. Eq (7) in [B] tries to make a selection so as to increase the angle between all pairs of vectors. [A] Clarkson, Kenneth L. "Coresets, sparse greedy approximation, and the Frank-Wolfe algorithm." ACM Transactions on Algorithms (TALG) 6.4 (2010): 1-30. [B] Aljundi, Rahaf, et al. "Gradient based sample selection for online continual learning." Advances in Neural Information Processing Systems. 2019.


Review 3

Summary and Contributions: A well known Theorem, a consequence of Caratheodory's convex hull theorem, states that if we have n functions over a probability measure on a large support, we can always choose a another measure over n+1 points in the support so that the expectation of the functions remains the same. The submission proposes two algorithms to find such a reduced support and measure that are incomparable, a deterministic one and a randomized one. THe submission includes experimental evaluation of the algorithms and some analysis.

Strengths: This is a very natural summarization problem and it is interesting to look at it algorithmically.

Weaknesses: The approaches (based on solving linear systems) seem basic and it is not clear if they offer new techniques. I would have liked to see some motivating discussion of potential applications. After feedback: Thank you authors for the feedback! My main concerns was getting more confirmation of novelly (natural problem, simple solution) but it was addressed by the feedback and other reviews. I increased my score. Still, an extended motivating discussion would be helpful .

Correctness: I did not verify but I did not see issues when spot-checking.

Clarity: It is written ok.

Relation to Prior Work: As far as I know.

Reproducibility: Yes

Additional Feedback:


Review 4

Summary and Contributions: This paper addresses the problem of reducing the support of discrete measures supported on N atoms to (n+1) atoms while preserving the expectations (integrals) over n given real-valued functions. It is built upon the Caratheodory theorem and illustrates its benefits on a fast least-square solver. The contributions are: 1) Theoretical conditions on the geometry of the n+1 atoms (theorem 3, negative cones). 2) A generic algorithm that uses this geometry (by maximizing the volumes of the cones) to reduce the support of any discrete measure. 3) The ability to combine this method with other existing ones.

Strengths: This paper is very relevant to the Neurips community since it tackles a well defined and well explained ML problem with a theoretically grounded method, which claims are proved and evaluated with care. I have also enjoyed the empirical validation on both real and synthetic data. In particular I have appreciated the analysis of the different regimes and the suggestion of the author to prefer this method for the big data regime and to combine it with previous work when the number of samples is low (the hybrid algorithm in the supplementary materials). The empirical validation also illustrates well the difference between the theoretical worst case complexity and the running time for real applications.

Weaknesses: I do not see real weaknesses in this paper.

Correctness: The claims are supported by proofs (mainly given in the supplementary materials) and I have empirically double checked theorem 3, mostly to better get the geometrical intuition behind the method. The empirical methodology seems solid.

Clarity: I am not an expert in this field but I have enjoyed reading this paper, since it is very well written and easy to follow. I have also appreciated the pedagogical effort of presenting the 2D case whenever needed to help the reader grasp the intuition.

Relation to Prior Work: This paper clearly discuss prior works, clearly states its contributions and also shows how it differs and how it can be combined with prior contributions.

Reproducibility: Yes

Additional Feedback: There is a small mistake on line 125, where it should be the negative cone of x^{\star} instead of x. I would recommend the authors to increase the font size and ticklabels size of all their figures. The labels are sometimes different from one figure to another, which does not help and the colors are sometimes changing for a given method from a figure to another. I would recommend keeping one color and one label per method across all figures. On line 230, the authors comment that the run time is approximately O(n). I believe that to illustrate this claim, it would be easier to plot figure 1 in log-log scale. On figure 3, the distributions of run times are totally concentrated on the left half of the plot and I would suggest to restrict the x axis to this range in order to better visualize the distributions. Also I believe the two rows could be merged into one. Showing the distribution for both the log-random and the randomized algorithm in the same plot.

[Author Response · NeurIPS 2020]

We thank all four reviewers for the careful reading and detailed feedback.

R1, *"Relation to previous work/bounds"*. The algorithms in [4, 5, 6] require to touch all $N$ points before the support
$x_1,...,x_{n+1}$ of $\hat{\mu}$ is found; in contrast, Algorithm 2 can finish much earlier by smart sampling; e.g. for $d = 2$, Algorithm
2 terminates after at most 2 steps (Theorem 4), which is a big reduction in the big data regime when $N \gg 2$; for $d > 2$,
the analysis is more subtle since pathological behaviour can occur with small probability, but generically the same
behaviour applies. We will add a table with runtimes to further highlight this point, which we believe to be a main
contribution of this work. The complexity bounds of Algorithm 1 give here only very conservative bounds for Algorithm
2; see also the discussion starting in line 152.

R1, *"Approximation?"*. Thank you for the suggestion! Firstly we should note that this would exclude important
applications, e.g. if $f_1,\ldots,f_n$ are monomials up to given degree, then $\hat{\mu}$ integrates every function exactly that is in
the linear span of $f_1,\ldots,f_n$, i.e. every polynomial up to given degree. This is for example essential to derive error
bounds for cubature, see [4, 5]. However, for other applications an approximation would suffice and we have thought
about this previously but, put simply, we do not know how one could calculate an approximation with significantly
less computational cost. The bottleneck remains in finding the support of $\hat{\mu}$. We believe approximate solutions are an
interesting open research question; in fact, already making precise what is meant by a "good approximation" allows for
many possibilities since one can choose many different metrics for measures to quantify when $\mu$ is close to $\hat{\mu}$ as well as
many different ways how an excess number $k > n+1$ of supporting atoms $\{x_1,\ldots,x_k\}$ should be penalized.

R1, *"choice of n-points random"*. In Algorithm 1, the points are simply drawn uniformly at random but not in Algorithm
2; for the latter only the first selection is random, and subsequent choices aim to maximize the volume of the (negative)
cone which is one of the main reasons for the performance of Algorithm 2. The intersection between the cone and the
negative cone is always empty (except for the origin) in both algorithms.

R1, *"Approximate Least Squares"*. We included all experiments done in [6] since it was a NeurIPS paper, thus provides
a natural and competitive benchmark. But yes, the measure reduction is in this application only advantageous if exact
solutions are required, which is often but not always the case, and we will add a sentence to note this.

R1, *"Complementary"*. In Appendix E we show how [4, 5, 6] can be combined with Algorithm 2. The hybrid
algorithm has a high chance of requiring very few steps to find the support, but for pathological cases it can rely on the
deterministic procedure of [4, 5, 6].

R1, *"Exposition/Pictures"*. R2, *"proofs in main text"*. Yes, we will add pictures and the extra page also allows to move
proofs to the main text and write more about the intuition. Figure 6 shows the situation in dimension 2 and we will
expand on this further.

R2, *"Reset strategy"*. The effects of the reset strategy can only be noticed when $N$ and $n$ are close, and $n$ is relatively
high. This is the reason why the lines are indistinguishable in the regime of Figure 1. However, the reset strategy
always guarantees the convergence of Algorithm 2 (in addition to being optimal in the sense discussed in line 188ff);
e.g. without it, a pathological case of cycling through a "sequence" of cone bases could occur, which is the reason why
we always apply it, even when $N$ and $n$ are not close or $n$ is relatively low.

R2, *"Explicit optimization"*. Thank you for the suggestions and references! We were not aware of [A, B] and we will
include both. [B] implicitly aims to maximize a volume, similar to what we try to achieve (see step 7-9 in Algorithm 2)
although we are directly motivated by polygon geometry going back at least to [*Sommerville, 1927, "The relations
connecting the angle-sums and volume of a polytope in space of n dimensions"*]. Ideally, we would like to maximize
directly the volume but this is a longstanding open question in discrete geometry, see e.g. [*Bollobas, 1997, "Volume
estimates and rapid mixing"*], or [15] for an overview; the proposed "angle maximization" in Algorithm 2 serves as a
computationally efficient proxy to this problem (although it is not perfect proxy; see pathological cases discussed in line
150ff and how to deal with them in line 250ff). [A] has a similar motivation, but works with gradients in continuous
space so it is not obvious to us how to apply it for the subset selection directly, however, combined with a continuous
relaxation we believe that this would be an interesting research question.

R3, *"basic techniques/applications"*. Yes, we combine fundamental results from algebra, discrete geometry, and
probability to derive a new algorithm. The approach of geometric sampling by exploiting cone geometry is very
different from previous approaches [4, 5, 6] to this classic problem; independent of this, even robustness (Proposition 3)
was not established for any of the previous algorithms [4, 5, 6]. As for applications, in addition to the ones done in
Section 4, and the related work cited in the introduction, we can refer to [5] to approximate integrals, while in [4] the
reduction procedure yields numerical methods for Stochastic Differential Equations and Partial Differential Equations.
The design and analysis of an effective measure reduction algorithm is the focus of this paper.

R4, *"figures/cone/log-log scale"*. Thank you for the kind words and remarks! Yes, it should be a negative cone; we will
change the figures as suggested. In line 230 we refer to Figure 2 (bottom) where the scale should already be log-log.

[Meta-Review · NeurIPS 2020]

All reviewers are in favor of acceptance, and I also recommend acceptance.